# Patient Specific Instruments and Patient Individual Implants—A Narrative Review

**DOI:** 10.3390/jpm13030426

**Published:** 2023-02-27

**Authors:** Christian Benignus, Peter Buschner, Malin Kristin Meier, Frauke Wilken, Johannes Rieger, Johannes Beckmann

**Affiliations:** 1Department of Traumatology and Orthopedic Surgery, Hospital Ludwigsburg, Posilipostr. 4, 71640 Ludwigsburg, Germany; 2Department of Orthopedic Surgery and Traumatology, Hospital Barmherzige Brüder Munich, Romanstr. 93, 80639 Munich, Germany; 3Department of Orthopedic Surgery and Traumatology, Inselspital, University Hospital Bern, University of Bern, Freiburgstr. 4, 3010 Bern, Switzerland

**Keywords:** custom-made implants, patient-specific implants, patient-specific instrumentation, Knee arthroplasty, hip arthroplasty, high-tibial osteotomy, kinematic alignment, total ankle arthroplasty, shoulder arthroplasty

## Abstract

Joint arthroplasties are one of the most frequently performed standard operations worldwide. Patient individual instruments and patient individual implants represent an innovation that must prove its usefulness in further studies. However, promising results are emerging. Those implants seem to be a benefit especially in revision situations. Most experience is available in the field of knee and hip arthroplasty. Patient-specific instruments for the shoulder and upper ankle are much less common. Patient individual implants combine individual cutting blocks and implants, while patient individual instruments solely use individual cutting blocks in combination with off-the-shelf implants. This review summarizes the current data regarding the implantation of individual implants and the use of individual instruments.

## 1. Introduction

Personalization in medicine is growing enormously and was introduced into orthopaedic surgery several decades ago. Interestingly, one of the first steps was the introduction of robotics in the field of arthroplasty. A large soft-tissue access was required for sufficient exposure. Due to this considerable disadvantage, robotics were banned, but they experienced a renaissance in the last decade [1]. Knee navigation systems were developed in arthroplasty towards the end of the 1990s with the assumption that the accuracy of the prosthesis fit would improve the survival rate of the prosthesis as well as clinical outcomes. The approach via CT-based navigation systems took place for the first time, with imageless systems evolving shortly after. Precision such as leg alignment could significantly be improved by the aid of navigation systems, however, clinical outcome was not. Actual robotic systems are somehow the combination of robots and navigation, again working either CT-based or imageless. Those systems are beyond the topic of this article [2]. In the further course, the broad acquisition of computed tomography (CT) data of bone surfaces was used to produce cutting blocks that would precisely guide the surgeon in the implantation of the prosthesis followed by individual prostheses [3].

These patient-specific implants and instrumentations were launched several years ago to facilitate and improve precise implantation, with the overall aim to improve the outcome of arthroplasty. On the one hand, there is the individual cutting block technology, which is referred to as patient-specific instruments or patient-specific instrumentations. Confusingly, the term patient-specific implants is also used, even though only the cutting blocks are custom-made and standard implants are used for implantation. These are to be distinguished from individual implants, which combine an individual cutting block technology together with individual implants, which can be found in the literature as true patient-specific implants or also as custom-fit or customized implants. Except for total knee arthroplasty, data concerning patient-specific instrumentations are rare, with results often being contradictory but promising. In the last few years, the results, particularly in precision, improved. This might also be attributable to improved scanning and printing technology. These techniques are increasingly used in osteotomies, ankle arthroplasty and shoulder arthroplasty as well as in knee arthroplasty with modern alignment philosophies. Higher costs must be charged up against reduced surgical time, blood loss and fluoroscopic time. Custom-made implants are primarily used, with promising results in hip and knee arthroplasty. Evidence, however, just shows the narrative advantage so far. These primary implants must still prove their effectiveness and superiority in long-term studies before widespread use can be recommended. A growing and clear indication for custom implants, however, is revision situations with bony defects or primary cases with bone deformity.

## 2. Knee Arthroplasty

In contrast to hip arthroplasty, a major problem in knee arthroplasty is the high number of patients who are not satisfied with the results of the operation. Postoperative pain and functional limitations often remain, which in the course of time may lead to prolonged physiotherapeutic measures or even reoperations. This represents a high socio-economic burden. Various factors play a decisive role in patient satisfaction, including the best possible restoration of patient anatomy. The implant design, the surgical technique and also the positioning or the alignment of the prosthesis is crucial in that context [4].

Patient-specific instrumentation (PSI) was introduced into knee arthroplasty roughly two decades ago and comprises the vast majority of the literature. Already, around 2015, there were several systematic reviews that showed no advantage over standard techniques with regard to component alignment as well as clinical outcome [5,6,7,8]. However, in the last few years, the results, particularly in precision, improved, which might also be attributable to improved scanning and printing technology. Furthermore, the accuracy of the produced instruments increased by using magnetic resonance imaging (MRI) data rather than CT data. Especially the remaining articular cartilage is hard to be estimated from CT reconstructions. Thus, the cutting blocks may not be able to make sufficient contact with the bony surface. MRI-based cutting blocks offer an easier and more accurate reconstruction in this context. The disadvantage of the MRI technique, however, is the higher susceptibility to motion artefacts. The costs and the extended examination time of the patients must also be considered [9,10]. Thienpont et al. demonstrated in a meta-analysis that the accuracy of femoral component alignment in the coronar plane as well as the global mechanical alignment were significantly improved by PSI. No differences were found with regard to alignment in the axial plane. However, the risk of poorer positioning and malalignment of the tibial component was approximately 30% higher with PSI than for standard instrumentation in both the coronal and sagittal planes [11]. Operative time and blood loss (regardless of calculating as blood volume or hemoglobin count) decreased with the use of the PSI technique compared to standard techniques, but these differences were minimal [11,12]. A more recent study from 2022 showed that tibial rotational positioning can be improved by PSI and that there are fewer outliers compared to conventional techniques [13,14]. Good results with few outliers were also shown for femoral rotational positioning when compared to conventional instrumentation. This is of paramount importance as an incorrect rotation of the femoral component affects the kinematics of the implanted knee prosthesis, possibly resulting in patellar tracking with anterior knee pain, instability and stiffness [15].

Regarding functional outcome, however, still no advantages were found in favor of PSI compared to conventional instrumentation [16,17]. Very interesting is the consideration of costs. A recent retrospective study in the US evaluated total hospital cost and readmission rate at 30, 60, 90, and 365 days in PSI-guided total knee arthroplasty (TKA) patients. The study matched 3358 TKAs with PSI with TKA-without-PSI patients. Mean total hospital costs were statistically significantly lower for TKA with PSI, at an astonishing USD 14,910 in the US medical system [18]. Another very interesting cost analysis study compared imageless robotics, image-based robotics, navigation and PSI in the medical system of Switzerland. The costs per case were lowest with navigation, comparable between imageless robotics and PSI at roughly USD 1500, and highest with image-based robotics by far.

The most important factors, linked to costs, were technical support and additional disposables. On the contrary, longer surgical times and additional surgical trays only had a minor effect on overall costs [19].

There are conflicting results regarding unicondylar arthroplasty, with each of three papers showing advantages in implantation accuracy [20,21,22] and no advantages in accuracy nor outcome [23,24,25], respectively.

With the recent “hot topic debate” of different alignment philosophies, PSI became the further impetus. The PSIs of modern technology could help to implement the plan of kinematic alignment or other novel alignment strategies more precisely. Again, data in the literature are sparse, but they show promising results for PSI with shorter operation times, as well as a lower number of instruments required, and therefore a possible simple and standardized solution for implementing kinematic alignment [26,27,28,29].

Individual, custom-made implants (CMI) have been available since 2006, with initially only one company (Conformis, Boston, MA, USA) launching unicondylar implants, which was then chronologically followed by bicompartmental, bicondylar cruciate ligament preserving, and most recently, posterior-stabilized bicondylar implants. A second company manufacturing individual implants has existed for a few years now (Symbios Orthopedie), producing only posterior-stabilized bicondylar implants to date. The main difference between both is the alignment based on the time of the manufacturer’s development. While Conformis is aiming for a neutral hip–knee–ankle axis with restoring asymmetry by an oblique joint line (since mechanical alignment was the gold standard in early 2000), Symbios allows a restricted alignment up to 3° in addition to an oblique joint line.

Two recent papers show that CMI have promising results in terms of fit, axis correction, more natural kinematics, patient satisfaction and cost neutrality [30,31]. The Orthopaedic Data Evaluation Panel (ODEP), as an advisory body to the National Health Service (NHS) in the UK, gave Conformis prostheses a 3A rating back in 2017. ODEP draws on data from the National Joint Registry (NJR) for England and Wales as well as expert opinions. Registry data showed a significantly lower early loosening rate for individual implants than for off-the shelf implants. The ODEP believes there is strong evidence of a substantial, patient-relevant improvement in clinical outcomes and a significant reduction in early loosening rates with the individual implant [30]. Meanwhile, the ODEP rating has reached a 7A rating.

On the other hand, neither Moret et al. [31], in a recent literature review, nor Müller et al. [32], in the most recent meta-analysis on total knee arthroplasty (TKA), could find a difference for the clinical outcome between conventional implants and CMI.

In another recent review, the implantation of individualized TKA is not even recommended. It did not demonstrate significant benefits in terms of knee and function scores or range of motion, and had higher early revision rates, although the latter were not statistically significant [33]. Demey et al. also failed to find any advantages in favor of individualized implants in a meta-analysis for partial joint replacement [34]. Higher rates of malpositioning, overcorrection, or loosening were also shown in one study each on TKA, bicompartmental knee arthroplasty (BKA), and unicondylar knee arthroplasty (UKA) [35,36,37]. However, the promising results of kinematic and biomechanical studies as well as patient-related outcome measurement (PROM) data from various case series suggest decisive improvements in clinical outcomes in favor of CMI [38].

Furthermore, there are three recent comparative studies on the products of both companies, which are mostly not included in meta-analyses. They show clear advantages of CMI compared to off-the shelf implants in terms of pain, mobility, overall outcome, and satisfaction for Conformis (iTotal^®^) [38,39,40] as well as Symbios (Origin^®^), with also very promising clinical and radiological results [41,42,43]. The latter comparative studies, however, might have conflicting bias as they are at least partly sponsored.

Worldwide, analogously, the number of knee revision surgeries is expected to increase enormously by 601% from 2005 to 2030, solely in the US. Multiple revisions often result in the difficult anchorage of components. Common options for dealing with reduced bone stock after revision surgery, trauma or tumor disease include bulk allografts, impaction grafts, metallic augmentation and porous metal cones/sleeves; however, there are situations where these options reach their limits. Here, CMI (even just the anchoring parts) are increasingly being considered [44]. However, high rates of re-revision occur compared to primary arthroplasty, with complication rates of up to 50% and survival rates of just about 54% after 8 years [45]. These data are based on case reports and small case series due to the inhomogeneity of the patient-specific remaining bone stock.

In summary, PSI shows mixed outcomes for alignment and positioning so far; however, the clear advantages are shorter operation time, reduced blood loss, as well as lower long-term costs. CMI still must prove its value, but the results are very promising.

## 3. Osteotomies

Osteotomies are performed with the aim to correct extra-articular deformities, particularly around the knee, as a pre-arthritic condition in symptomatic patients. The correct analysis of deformities is crucial [46]. Multiplanar deformities exist and are not rare, making either bifocal osteotomies or multiplanar osteotomies necessary, e.g., for the tibia, not just coronal but also sagittal planes (slope) have to be considered.

For this, the angle of correction as well as the sawblade direction are essential.

For preoperative planning, a weight-bearing coronar X-ray of the knee is taken to determine the corrective coronal-plane angle, the size of the osteotomy gap and, if necessary, the screw length [47]. Additionally, a lower-leg X-ray is needed, when multiplanar corrections with additional slope correction have to be addressed.

The standardized positioning of the leg during preoperative and intraoperative X-ray diagnostics is crucial but prone to failure. Measured angles and the range of correction may differ enormously as a result. Likewise, a biplanar correction is difficult to depict with the two-dimensional X-ray procedure and constitutes a further source of error [48]. Here, PSI could clearly assist, being less prone to such failures. However, PSI was introduced to help in several aspects. It can also be used to determine the length and thickness of the plate as well as the length of the necessary screws. This can be prepared preoperatively and thus leads in consequence to a reduction in operation time. The fluoroscopic time can also be reduced compared to conventional osteotomies and the desired correction can be achieved well with the help of PSI [49]. Furthermore, a short learning curve for optimizing an open-wedge high tibial osteotomy using PSI could be demonstrated. The evaluation of the learning curve already showed an advantage in terms of operating time in the first learning phase of the surgeons. In the stable plateau phase of the learning curve, a potential reduction of the operating time to approximately 70% can be assumed compared to the conventional technique [50]. Although good results of the leg axis were shown, there was no significant clinical improvement compared to conventional osteotomies [49,50,51]. The procedure using PSI also seems to be safe in patients with a pre-operated knee joint. Here, a common previous ACL reconstruction should be mentioned. When planning the osteotomy, the position of the former ACL-drill channels must be taken into account, as well as the hardware inserted. It is essential to avoid the weakening of the inserted ACL reconstruction through the incorrect positioning of the plate or incision [52]. A recent systematic review (of Level-III and -IV studies, however) could confirm a highly accurate coronal plane alignment with a low rate of outliers, significantly shorter operative times and decreased intraoperative fluoroscopy when compared to conventional techniques for both distal femoral as well as proximal tibial osteotomies [53]. Therefore, PSI seems to be a reliable option to facilitate osteotomies and a possible option for pre-operated patients or patients with anatomical norm variants as well. On the other hand, the higher costs of PSI must be weighed up against reduced surgical and fluoroscopic time.

Patient-specific implants obviously have no major role in osteotomies, with well-established plates on the market.

## 4. Shoulder Arthroplasty

In recent years, progress in shoulder arthroplasty has focused in particular on the development of PSI and the further development of inverse shoulder arthroplasty implants and glenoid components, which have gained enormous popularity.

The placement of the glenoid component is often technically challenging and especially difficult in patients who already have significant bone loss at the glenoid due to severe osteoarthritis [54]. Glenoid deformities, as biconcave, retroverted glenoids with humeral subluxation, can often lead to increased complication rates after the implantation of an anatomic prosthesis [55], which is why the implantation of a reverse shoulder prosthesis is often performed in these cases [56]. To better assess the anatomy preoperatively, CT scans are usually performed, from which PSI can also be made. In this way, a target instrument for the glenoid can be manufactured preoperatively, whereby attention must be paid to several parameters such as centering, inclination, anchoring in the bone, and the subluxation of the humeral head [57]. A 2018 meta-analysis of glenoid component implantation in cadavers and humans, comprising 12 studies, showed that deviation from the preoperative planning was significantly lower for the version, inclination and entry point of the pin using PSI compared to standard implants. Furthermore, outliers with a deviation > 10° or 4 mm were significantly decreased by PSI (15.3% vs. 68.6%) [58]. However, another meta-analysis from 2019 failed to detect a significant difference between the PSI group and standard implants in terms of version error, inclination error or positional offset. This study described that PSI are expensive to manufacture and take about 6 weeks to be delivered, but they seem to be justified in complicated cases nevertheless [57]. As outsourcing PSI production to external companies is associated with long delivery times and high costs, another study described the use of 3D printers that allow on-site production. The PSI group delivered reliable results; however, only a small case series of cadavers was comprised [59].

Patient-specific implants are not (or not yet?) used in primary arthroplasty but are a good option for patients with complex cases, especially in tumor surgery when large bone resections have to be addressed [60].

However, with the increasing number of primary implantations of artificial shoulder joints, the number of revision operations is also steadily rising. Glenoid loosening and instability of the prosthesis are the most frequent reasons for revision [61]. Due to the pronounced bone loss in the case of replacement operations, the anchoring of the revision prosthesis can be significantly more difficult. Therefore, the need for individual solution strategies in the form of custom-made implants increases. For these cases, some producers offer the production of individual implants from the 3D printer based on 3D-CT or MRI data. Due to the high production costs, however, this is used more individually [57].

In conclusion, the results of the lower deviation in PSI are promising, but the technique is still costly and time-consuming and therefore only considered in individual cases.

## 5. Hip Arthroplasty

PSI in hip surgery will possibly gain influence with osteotomies and have already been introduced into arthroplasty by the guidance of femoral resection as well as cup orientation.

The data concerning custom implants in total hip arthroplasty (THA) are very limited so far and gather around few research centers; however, they have very good results overall. Multicenter, randomized controlled trials and registry data would be desirable to be able to confirm the evidence of the results across the board. Custom implants have been introduced into THA more than two decades ago. Presumably because of the outcome of THA being by far better than in TKA, the manufacturing of customized implants seems to be mainly for special anatomies. It may be especially beneficial for young patients with dysplastic hips. In those patients, standard implants are difficult to implant, but good activity and long survival rates are needed. Only Hitz et al. found a revision rate of 23.1% (six cases) in higher grade dysplasia with, however, good survival rates in terms of the loosening of the stem and cup [62]. Jacquet et al. showed a survival rate of 96.8% after a long-term follow-up of 20 years in a group of patients younger than 50 years and a 96.1% survival rate in those with high-grade developmental dysplasia of the hip, all with good clinical results [63].

The implantation of custom-made cementless stems also seems to be useful after the fusion of the hip joint, with an excellent survival rate and results after 15 years. Flecher et al. examined 23 patients who underwent conversion from a fused hip to THA with a custom femoral implant. Overall, the postoperative complication rate was 26%, which is in line with the literature in this special and rare patient population and included especially heterotopic ossification and aseptic loosening. Conversely, the rate of intra-operative complication was very low, e.g., no intra-operative fracture was observed. It is hypothesized that the use of custom protheses, designed to fit perfectly with the intramedullary anatomy, may explain those differences [64].

In the case of large acetabular bone defects, which are frequently encountered in revision arthroplasty and an enormously growing problem due to increasing numbers of arthroplasties and demographic development itself, standard implants are often inadequate. Bone defects of the acetabulum can be classified according to Paprosky [65] or the American Academy of Orthopedic Surgeons (AAOS), for example [66]. The AAOS classification distinguishes between four different degrees of severity (type I to IV), while Paprosky differentiates six defect types (type I, IIa, IIb, IIc, IIIa, IIIb). If possible, it is better to “down-grade” the defect by means of the biological reconstruction of the acetabular bone. Especially in young, active patients, this can significantly simplify any revision surgery that may occur later. However, the possibility of biological reconstruction is often not sufficient. The overall goal is to restore the center of rotation as well as stability at the acetabular component. At least 50% of the surface of the cementless implant should be covered with autochthonous bone. Types IIIa and IIIb, according to Paprosky, as well as defects according to AAOS types III and IV, are acetabular defects for which different treatment regimens are available with “Jumbo”-cups, pedestal cups or modular options with special augments. Surgical “easiness” as well as defect size caused the desire for a stable monobloc implant that enables defect bridging. This led to the development of individual partial pelvic replacements, especially for the higher-grade defects that are usually associated with instability. The proportion of so-called “mega defects” in acetabular revision cases is given as 1–5% [67]. The available studies in the literature are difficult to compare because the patients’ initial situations, prosthesis design and classification of the defects often differ significantly, as does the philosophy of how to reconstruct the defect. Scheele et al. recommend an individual partial pelvic replacement for bone defects that exceed the incisura ischiadica, a non-constructible dorsal rim or pelvic discontinuity [68]. Chiarlone et al. analyzed custom-made implants for large bone defects of the acetabulum in revision total hip arthroplasty in a systematic review and included 634 custom-made acetabular implants (627 patients), with a mean follow-up of 58.6 ± 29.8 months from 18 studies. Good clinical and functional results were seen together with a survival rate of 94.0 ± 5.0%. Despite this, the re-operation rate was as high as 19.3 ± 17.3% and the mean complication rate was 29.0 ± 16.0%, with instability being the most common complication [69]. The disadvantage is the high cost of these often-huge custom implants, so they should be used only in special cases, where modular implants cannot be used. The factor time is also important due to the ordering and manufacturing of the implants taking several weeks, during which changes in the patient’s bone situation may occur [67]. As has been demonstrated in this paper, custom-made implants show promising clinical results, but considering high costs and long production times, their use has to be judged carefully in every case.

## 6. Total Ankle Arthroplasty (TAA)

The data on PSI TAA are very sparse. There are currently three different types of implants for PSI TAA, two of which are component designs (one talar and one tibial component) and one is a three-component system with a mobile bearing. In a cadaveric study, PSI positioned the implants to less than 2° in all rotational and translational degrees of freedom [70].

Posttraumatic deformities as well as ligament injuries and previous surgeries can make alignment correction more complicated and less predictable. This contrasts with nontraumatic osteoarthritis. Albagli et al. [71] compared the clinical and radiological outcomes of patients with end-stage arthritis—traumatic versus nontraumatic—treated with an implant with CT-guided patient-specific preoperative plans and patient-specific incision patterns. In contrast to previous studies on patients with total ankle arthroplasty in posttraumatic patients, it was shown that there was no difference in patient satisfaction, short-term clinical outcome and radiological outcome when using CT-guided preoperative plans and incision patterns compared to nontraumatic patients. In several studies, the accuracy of implant positioning between the PSI groups and the standard implants was comparable, with no superiority of one group. Patient-specific templates enabled the reproducible positioning of the tibial implant in more than half of the cases, compared to preoperative planning. Discrepancies occurred mainly in severe preoperative varus deformities. In these cases, there are certainly also difficulties in conventional surgery. Postoperative alignment also showed comparable results. The studies were each conducted with experienced surgeons. To what extent an influence exists with inexperienced surgeons could not be shown here [71,72,73,74,75].

The complication and revision rates were comparable after both PSI TAA and the implantation of standard implants [76]. Additionally, the implant size of the tibial component could be estimated quite well using PSI TAA. However, the estimation of the talar component often showed poor results, sometimes less than 50% [76].

After a short follow-up, PSI TAA, using fixed-bearing CT-guided patient-specific implants, showed good results in both traumatic and nontraumatic arthritis compared to standard implants [71]. These results differ from traditional beliefs regarding poorer results with total ankle arthroplasty in posttraumatic patients. Again, surgical time has been shown to be shorter with PSI TAA [74,77], and fluoroscopic time can also be significantly reduced [74]. One study identified a reduction in cost in the PSI group, but this could only be attributed to the reduced surgical time [77]. Further studies with more patients and a longer followup are needed to demonstrate the benefits and theoretical advantages of PSI in TAA.

To date, there have been no studies using patient-specific implants.

## 7. Conclusions

Except for TKA, which is the focus of many studies, data concerning PSI are rare for other indications, with results being contradictory but promising. In the last few years, the results, particularly in precision, have improved, which might also be attributable to improved scanning and printing technology. The usage in osteotomies, ankle arthroplasty and shoulder arthroplasty is growing, which is also true in knee arthroplasty with modern alignment philosophies, which are—talking about kinematic alignment—mostly a compromise of restoring individual anatomy and using symmetric, non-individual implants. Higher costs have to be charged up against reduced surgical time, blood loss and fluoroscopic time. Custom-made implants are primarily used with promising results in hip and knee arthroplasty. The evidence, however, simply shows the narrative advantage so far. These primary implants must still prove their effectiveness and possible superiority in long-term studies before widespread use can be recommended. A growing and clear indication for custom implants, however, is revision situations with bone defects.

## Data Availability

Not applicable.

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
