# Peer review of "Patient Specific Instruments and Patient Individual Implants—A Narrative Review"

_jpm, 2023, doi:10.3390/jpm13030426_

Round 1
Reviewer 1 Report
Well presented and very balanced review of the “personalized surgery” approach in the field.
The manuscript is very informative, detailed and well structured.
I recommend it for publishing in present form.
Author Response
Well presented and very balanced review of the “personalized surgery” approach in the field.
The manuscript is very informative, detailed and well structured.
I recommend it for publishing in present form.
We thank for this encouraging statement.
Reviewer 2 Report
Interesting and timely topic with current controversies to explore.
Where are the cost-savings achieved? Can descibe in more detail.
Needs a fair amount of editing for language, grammer and conciseness.
Author Response
Interesting and timely topic with current controversies to explore.
Where are the cost-savings achieved? Can descibe in more detail.
Needs a fair amount of editing for language, grammer and conciseness.
We apologize and had the manuscript edited by a native English speaker.
Concerning cost-savings, we added some sentences. Please also find the new Figure which summarizes nicely all subtopics.

Reviewer 3 Report
The proposed narrative review manuscript is interesting. However, the presentation if fully "narrative". It would be better to present a table for supporting the conclusion.
Author Response
The proposed narrative review manuscript is interesting. However, the presentation if fully "narrative". It would be better to present a table for supporting the conclusion.
We added a Figure which summarizes the conclusion and main facts for better visualization.

Reviewer 4 Report
This is a narrative review on current data regarding the implantation of individual implants and the use of individual instruments.
However, the topic is various from knee arthroplasty to ankle arthroplasty widely. There is no hot problem in this paper. In addition, the summarizes in each section is short of clinical meaning.
Author Response
This is a narrative review on current data regarding the implantation of individual implants and the use of individual instruments.
However, the topic is various from knee arthroplasty to ankle arthroplasty widely. There is no hot problem in this paper. In addition, the summarizes in each section is short of clinical meaning.
Thanks for your evaluation. Being a narrative review a hot problem was beyond the topic of this manuscript which was meant to give an overview and therefore fit perfectly into the Special issue.
We adopted the summarizing slightly. Please also find the new Figure.
The new Figure gives a visual summary.

Round 2
Reviewer 3 Report
The current version may be considered for publication.
Reviewer 4 Report
Accepted